# Comparative Transcriptome Analysis of Two Sweet Sorghum Genotypes with Different Salt Tolerance Abilities to Reveal the Mechanism of Salt Tolerance

**DOI:** 10.3390/ijms23042272

**Published:** 2022-02-18

**Authors:** Chengxuan Chen, Xiaoling Shang, Meiyu Sun, Sanyuan Tang, Aimal Khan, Dan Zhang, Hongdong Yan, Yanxi Jiang, Feifei Yu, Yaorong Wu, Qi Xie

**Affiliations:** 1State Key Laboratory of Plant Genomics, Institute of Genetics and Developmental Biology, The Innovative Academy of Seed Design, Chinese Academy of Sciences, Beijing 100101, China; chenchengxuan@genetics.ac.cn (C.C.); shang8011@126.com (X.S.); sunmeiyu@ibcas.ac.cn (M.S.); sytang@genetics.ac.cn (S.T.); aimal_khan_p@yahoo.com (A.K.); dzhang@genetics.ac.cn (D.Z.); ffyu@genetics.ac.cn (F.Y.); yrwu@genetics.ac.cn (Y.W.); 2University of Chinese Academy of Sciences, Beijing 100049, China; 3Crop Resources Institute, Heilongjiang Academy of Agricultural Sciences, Harbin 150080, China; hljcrop@126.com (H.Y.); jyxiheb2009@163.com (Y.J.)

**Keywords:** comparative transcriptome, salt stress, sorghum, transcription factors

## Abstract

Sweet sorghum is a C4 crop that can be grown for silage forage, fiber, syrup and fuel production. It is generally considered a salt-tolerant plant. However, the salt tolerance ability varies among genotypes, and the mechanism is not well known. To further uncover the salt tolerance mechanism, we performed comparative transcriptome analysis with RNA samples in two sweet sorghum genotypes showing different salt tolerance abilities (salt-tolerant line RIO and salt-sensitive line SN005) upon salt treatment. These response processes mainly focused on secondary metabolism, hormone signaling and stress response. The expression pattern cluster analysis showed that RIO-specific response genes were significantly enriched in the categories related to secondary metabolic pathways. GO enrichment analysis indicated that RIO responded earlier than SN005 in the 2 h after treatment. In addition, we identified more transcription factors (TFs) in RIO than SN005 that were specifically expressed differently in the first 2 h of salt treatment, and the pattern of TF change was obviously different. These results indicate that an early response in secondary metabolism might be essential for salt tolerance in sweet sorghum. In conclusion, we found that an early response, especially in secondary metabolism and hormone signaling, might be essential for salt tolerance in sweet sorghum.

## 1. Introduction

Soil salinity is one of the major factors leading to soil degeneration and desertification in the worldwide natural environment. Salt stress is a common abiotic stress that compromises crop growth in arable land [1]. Under salt stress, the osmotic pressure of the soil solution is high, which restricts plant growth and reduces the ability to take up nutrients and water, leading to osmotic stress. Meanwhile, the accumulation of Na^+^ in plant tissues leads to ionic toxicity [2,3]. The mechanism of plants to respond and adapt to salt stress is complex. Many biological processes in plants are affected under salt stress, such as photosynthesis, ion balance, energy metabolism, secondary metabolism, and protein synthesis [4,5]. The transcriptional level of many genes changed in these processes [6,7].

Sorghum (*Sorghum bicolor* (L.) Moench) is a C4 plant which originated in Africa [8]. Due to its evolution to adapt to the hot semi-arid environment of Africa, some varieties have a good ability to tolerate abiotic stress, adapting to growth in semiarid and arid regions [9]. Sweet sorghum contains varieties that accumulate soluble sugars in the stalk. It has a fast growth rate and large biomass, and can be used for forage, silage, fiber, fuel, and syrup production [10,11]. Some salt-tolerant varieties have been cultivated on saline-alkali land [12]. In previous studies, it has been found that salt strongly inhibits the germination and seedling growth of sorghum [13,14,15]. However, the variation in salt tolerance ability is large between sorghum genotypes [16,17]. Several quantitative trait loci (QTLs) related to salt tolerance at the germination and seedling stages were identified by linkage analysis [15].

In addition to physiological or genomic levels, studies at the transcriptional level can also explain the salt tolerance mechanism of sorghum. Next-generation sequencing technology (NGS) is an efficient tool for transcriptomics studies [18]. High-throughput RNA-sequencing (RNA-seq) analysis has been used to monitor gene expression in response to salt stress in many plant species [6,19,20]. Several transcriptomics studies by microarrays or RNA-seq have been carried out on the osmotic stress response [19,20,21,22,23,24,25,26,27,28]. However, most of these studies focused on one or two time points, representing only a snapshot of the transcriptional changes were captured, as the response to abiotic stress is a continuously dynamic process in plants [19,20,21,22,23,24,25,26,27,28]. In this study, we used RNA-seq technology to compare the dynamic transcriptional changes of a salt-tolerant inbred line (RIO) and a salt-sensitive inbred line (SN005) in sweet sorghum under salt stress. Our results provide further insight into the dynamic and complex regulatory networks in response to salt stress in sorghum. We aimed to identify the mechanisms and key genes involved in the salt stress response and their regulatory crosstalk network.

## 2. Results

### 2.1. Phenotypic and Physiological Responses to Salt Stress of Two Sweet Sorghum Inbred Lines

Two sorghum lines, RIO and SN005, were chosen and analysed in detail based on the screen of numbers of sweet sorghum inbred lines due to their difference in salt tolerance ability. At the germination stage, salt stress significantly inhibited the germination of SN005. The germination ratio of SN005 decreased to 87.8% under the 100 mM NaCl treatment, while it had almost no effect on RIO (Appendix A). By increasing the salt concentration, the germination ratio of SN005 decreased dramatically to 50% in 200 mM and 16.7% in 300 mM NaCl solution, while RIO maintained 86% and 71%, respectively (Appendix A), suggesting the higher sensitivity of SN005 than RIO under high NaCl concentrations at the germination stage. Furthermore, we checked the salt tolerance ability of two lines at the seedling stage and treated them at the three-leaf stage with four concentration gradients of NaCl (0, 150, 200, and 250 mM). After the 7-day treatment, the growth of the two genotypes was inhibited, but compared with SN005, the RIO seedlings grew better and remained green when treated with various concentrations of NaCl (Figure 1A).

The relative dry weights of RIO were 0.96 and 0.91 under 150 and 200 mM NaCl treatments, while the weights of SN005 were 0.89 and 0.76, respectively. The growth of SN005 was inhibited significantly in 200 and 250 mM NaCl solutions (Figure 1B). We also tested the influence of photosynthesis under salt treatment. As shown in Figure 1C, the actual PSII efficiency (ΦPSII) decreased dramatically in SN005 with increased NaCl concentrations compared to RIO. Under 250 mM NaCl solution, ΦPSII decreased 70.1% in SN005, while it only decreased 28.2% in RIO. Meanwhile, the relative chlorophyll content (SPAD) decreased in both lines after 7 days of salt treatment. The chlorophyll content decreased 33.8%, 44% and 45.3% under 150, 200 and 250 mM NaCl solutions in RIO and 39.8%, 59.9% and 71.2% in SN005. Although the chlorophyll content was reduced in both lines under salt treatment, there were no significant difference in the chlorophyll content in RIO between different salt concentration treatments. In contrast, the chlorophyll content declined rapidly as the salt concentration increased in SN005 (Figure 1D). These results showed the stronger adaptability of RIO to salt stress compared with SN005. Based on these phenotypic and physiological results, we chose RIO as the salt-tolerant line and SN005 as the salt-sensitive line for the transcriptome study.

### 2.2. Transcriptome Sequencing and Differentially Expressed Gene Analysis

The shoot tissues from two genotypes before salt treatment (CK) and 1, 2, 6, 12, and 24 h after salt treatment were collected for transcriptome sequencing. After filtration, a total of 226.5 billion nucleotides and 1.5 billion clean reads were generated. More than 93% of reads reached the Q30 level. The mapping rate of each sample was more than 91% (Appendix A).

Differentially expressed genes (DEGs) were identified at different time points after salt treatment compared to the control (Figure 2). In total, 5823 DEGs were identified in RIO and 5658 in SN005. Among them, 3460 genes were differentially expressed in both genotypes at least at one time point. In RIO, 1437, 1591, 2784, 2489, and 2078 DEGs were identified at 1 h, 2 h, 6 h, 12 h, and 24 h, respectively. In SN005, 1232, 1012, 2373, 2300, and 2701 DEGs were identified at 1 h, 2 h, 6 h, 12 h, and 24 h, respectively (Figure 2A).

Most DEGs were detected after 6 h of treatment in both lines, indicating that a wide response to salt stress occurs after this time point (Figure 2A). Notably, at the early stage of treatment (1 h and 2 h), more genes were differentially expressed in the salt-tolerance line RIO than in the salt-sensitive line SN005. The number of overlapping DEGs between different genotypes showed that a large number of overlapping DEGs appeared at 6 h (1364) and 12 h (1349) after treatment, while fewer overlapping DEGs appeared at 1 h (394) and 2 h (472) (Figure 2B).

Based on these data, we found that the number of responsive genes in RIO was larger than that in SN005 at 1 h, 2 h, 6 h, and 12 h of treatment but lower than that in SN005 at 24 h. For example, the number of DEGs in the salt-tolerant line RIO reached a maximum of 2784 at 6 h, while the salt-sensitive line SN005 reached a maximum of 2701 at 24 h (Figure 2A). Meanwhile, fewer overlapping DEGs at early stages (1 h and 2 h) reflected the differences in early response between genotypes. Fewer overlapping DEGs also appeared at 24 h (873), suggesting that the physiological statuses may be different between genotypes at a later stage (Figure 2B). After salt treatment, the dynamic change in gene expression revealed that the tolerant line responded more quickly and restored homeostasis earlier than the sensitive line. Previous research in abiotic stress RNA-seq analysis studies also discovered a similar phenomenon: stress-tolerant lines usually rapidly respond at the transcript level under stress [29,30].

### 2.3. Gene Ontology Enrichment Analysis of DEGs

Gene ontology (GO) enrichment analysis of DEGs at different time points was performed in two genotypes. The dynamic profiles of 26 significant enrichment GO terms are presented (Figure 3). The significantly enriched GO terms of biological process (levels 1–6) in at least one sample were selected. The dynamic profiles of GO terms revealed the difference in response to salt stress between the salt-tolerant line RIO and salt-sensitive line SN005. The categories “Cell Cycle” and “Chromatin Organization” were significantly enriched in SN005 at 1 h and 24 h after treatment. This revealed the influence of salt stimulus on cell division in the salt-sensitive line. In contrast, the categories related to the stress response, such as “Response to Stress”, “Response to Abiotic Stimulus”, “Response to Osmotic Stress”, “Oxidation Reduction” and “Protein Folding”, were enriched significantly at 1 h in the salt-tolerant line RIO, which indicated an efficient early response strategy in the tolerant line. Interestingly, DEGs at 1 h mainly focused on the categories of secondary metabolic process, such as “Phenylpropanoid Metabolic Process”, “Flavonoid Metabolic Process”, “Pigment Metabolic Process”, “Vitamin E Biosynthetic Process” and “Lignin Metabolic Process”, and hormone signaling pathways, such as “Response to Abscisic Acid Stimulus”, “Response to Jasmonic Acid Stimulus” and “Response to Salicylic Acid Stimulus”. These results suggested that these processes may participate in the early response to salt stress and lead to salt tolerance in RIO.

### 2.4. Gene Expression Pattern Cluster Analysis

In the time course transcriptome analysis, circadian-related genes were also identified as DEGs that were interference factors [31]. In addition, common response mechanisms for salt stress existed in the two genotypes [32]. These genes usually presented similar expression patterns among the two genotypes. We used expression pattern cluster analysis to divide the common response DEGs and genotype-specific response DEGs at the dynamic expression level to exclude these DEGs. Gene expression pattern cluster analyses were performed by constructing co-expression network in the weighted correlation network analysis (WGCNA) R package [33]. DEGs with mean FPKM ≥ 1 at least at two time points were selected. In total, 6116 genes (4541 DEGs in RIO and 4297 DEGs in SN005) were analysed. The expression patterns of these genes from the two genotypes were labeled by source and combined together for cluster analysis. Genes with similar expression patterns were clustered in the same WGCNA modules. These DEGs were divided into 22 co-expression modules (Appendix A). Among them, 2194 genes presented a similar expression pattern in the two genotypes because their expression pattern from the two genotypes clustered in the same co-expression module. Meanwhile, 2856 DEGs in RIO and 2560 DEGs in SN005 presented different expression patterns between genotypes (Figure 4A). Based on the cluster results, DEGs were clustered into three gene sets: common response genes, RIO-specific response genes and SN005-specific response genes.

GO enrichment analysis was performed for three gene sets. In the common response gene sets, genes were enriched in categories related to stress response, such as “Response to Abiotic Stimulus”, “Response to Oxygen-Containing Compound” and “Response to Stress”. DEGs in metabolite pathways, such as polysaccharide, lipid and secondary metabolite, and hormone signaling pathways were also enriched in this set (Figure 4B, Appendix A). In the expression pattern clusters, 39.8% of common response genes were clustered in WGCNA module ME17. Genes in ME17 were differentially expressed from 1 h to 6 h of treatment (Appendix A). The most common responses occurred in the first 6 h and were different in the later stage between the two genotypes.

In the RIO-specific response gene set, consistent with the GO enrichment analysis (Figure 3), genes were significantly enriched in the categories related to secondary metabolites processes, such as “Pigment Metabolic Process” and “Isoprenoid Metabolic Process” (Figure 4C, Appendix A). The GO term “Ion Transport” was also significantly enriched in the RIO-specific response gene set. These results revealed that ion transport genes might contribute to salt stress resistance in RIO (Figure 4C, Appendix A). In contrast, salt-sensitive line SN005-specific response genes were significantly enriched in categories related to stress categories, such as “Oxidation-Reduction Process”, “Response to Abiotic Stimulus” and “Response to Water Deprivation”. These results revealed that homeostasis was disrupted in SN005 plants under the failed resistance strategy (Figure 4D, Appendix A).

### 2.5. Secondary Metabolic Process

The secondary metabolic process is an important part of antioxidative systems. Many studies have found in vitro antioxidant activity in some secondary metabolites, such as flavonoids, anthocyanin, phenolic acids, sesquiterpenes, and coumarins, that play an important role in plant stress tolerance (Bian et al., 2019). In addition, secondary metabolites also contribute to osmoregulation and protection of hydrophilic cellular components under stress conditions (Bartwal et al., 2013). In secondary metabolites, the phenylpropanoid metabolic pathway is essential because it participates in the synthesis of many secondary chemicals, such as flavonoids, anthocyanins, tannins, coumarins, and lignins (Vogt, 2010). In GO enrichment analysis, genes in the secondary metabolic process were enriched earlier and more significantly in the salt-tolerant line than in the salt-sensitive line (Figure 3). In our study, 190 DEGs involved in the secondary metabolic process were identified at least at 1 time point. Sixty-four of them were identified as common response genes. Sixty-eight of them were identified as RIO-specific response genes, while sixty of them were identified as SN005-specific response genes (Appendix A). Among these DEGs, 114 genes were involved in the phenylpropanoid metabolic process.

GO enrichment analysis showed that there was a significant difference between genotypes at the first hour after treatment. Seventy-five DEGs in RIO, involved in the secondary metabolic process, were differentially expressed at 1 h, while thirty-six were differentially expressed in SN005. Forty-one of them were differentially expressed only in RIO at 1 h but not in SN005. For example, a putative *Chalcone synthase* (*CHS*) gene, *Sobic.005G137300*, encoding a key enzyme in flavonoid biosynthesis [34], was significantly upregulated under treatment in RIO but not SN005. Two *Cinnamate-4-hydroxylase* (*C4H*) genes, encoding a monooxygenase enzyme in the phenylpropanoid pathway [34], were significantly upregulated at 1 h after treatment in RIO but not SN005. Six *4-coumaroyl CoA ligase* (*4CL*) genes were identified in sorghum genome. 4-coumaroyl CoA ligase is a key enzyme that participates in the last step of the phenylpropanoid pathway [35]. Two *4CL* genes were differentially expressed in both genotypes, and four others were differentially expressed only in RIO. Among them, two *4CL* genes were significantly upregulated at 1 h after treatment in RIO but not in SN005. An anthocyanin 3′-O-beta-glucosyltransferase (*3GT*, *Sobic.002G366500*) was significantly upregulated after 1 h only in RIO. An anthocyanins dihydroflavonol 4-reductase (*DFR*, *Sobic.003G230900*) gene, a cinnamyl alcohol dehydrogenase 1 gene (*CAD1*, *Sobic.010G071800*), and a 6-phosphogluconolactonase 1 (*PGL1*, *Sobic.002G368000*) gene were identified as RIO-specific response genes that were significantly upregulated only in RIO (Appendix A). In brief, secondary metabolism-related genes quickly responded in the tolerant genotype, which might explain the differences in salt tolerance ability between the two genotypes. It also revealed that antioxidative secondary metabolites might play an important role in the salt stress response in sweet sorghum.

### 2.6. Hormone Signaling

The GO enrichment analysis also identified a significant difference in hormone signaling pathways between the two genotypes (Figure 3). It has been established that hormone signaling pathways play important roles in the salt stress response [36,37]. GO enrichment analysis showed that DEGs in response to abscisic acid (ABA), jasmonic acid (JA) and salicylic acid (SA) signaling pathways were significantly enriched at 1 h after treatment in the salt-tolerant line RIO (Figure 3). Our results identified 60 hormones signaling response DEGs differentially expressed at 1 h in RIO and only 35 DEGs in SN005. Among them, 42 DEGs were differentially expressed only in RIO. For example, three *Jasmonate-zim-domain proteins* (*JAZs*, *Sobic.001G482700, Sobic.002G214800, Sobic.006G056400*) were significantly upregulated within 1 h in RIO but not in SN005. An aquaporin gene, *Sobic.002G125700*, as a homologous gene of *AtPIP2B*, was upregulated at 1 h only in RIO. Five MYB family transcription factors and one NAC family transcription factor, which are related to hormone signaling, had similar differences between genotypes (Appendix A). The rapid response to stress-related hormones might also explain the strong tolerance ability of RIO.

### 2.7. Differential Expression Analysis of Transcription Factors

Previous studies have demonstrated the important roles of transcription factors in environmental stress responses in plants [38,39,40,41]. The differences in the expression patterns of transcription factors can reflect the different responses of two genotypes under salt stress. In the Plant Transcription Factor Database (PlantTFDB), 1860 transcription factors (TFs) were identified in the sorghum reference genome and classified into 57 TF families [42]. In our study, 349 TFs in RIO and 354 TFs in SN005 were differentially expressed. Among them, 208 TFs were differentially expressed in both genotypes. These DE TFs belonged to 45 families, including several key regulatory TF families involved in the response to abiotic stress [38], such as bHLH, MYB, ERF, WRKY, bZIP and NAC (Figure 5, Appendix A). However, the two genotypes showed apparent differences in DE TF families. The three TF families with the largest number in RIO were bHLH (34), MYB (27), and ERF (26), while they were ERF (27), MYB (26), and MYB-related (25) in SN005 (Figure 5A,B). For example, *Sobic.001G473900* is an ERF-type TF that was upregulated at 24 h after salt treatment only in RIO (Appendix A). The homologous genes of *Sobic.001G473900* in Arabidopsis responded to abiotic stress by ethylene-mediated pathways [43]. *Sobic.003G007700* is a homologous gene of *AtbHLH68* that has a function in the response to abiotic stress through an ABA-dependent pathway [44]. It was upregulated at 24 h after salt treatment only in RIO (Appendix A). *Sobic.007G062200* is another bHLH-type TF that was upregulated at 24 h after salt treatment only in RIO (Appendix A). The homologous genes of *Sobic.007G062200* in rice and Arabidopsis enhance stress tolerance by regulating proline biosynthesis and ROS scavenging pathways [45,46]. As the key regulatory functions for transcription factors, the difference in DE TFs revealed the different strategies for salt stress response between the two genotypes.

Early responsive transcription factors are involved in early signal transduction to activate stress responses and the regulation of stress-related genes, eventually resulting in plant stress tolerance [4]. In this study, 155 TFs in RIO and 113 in SN005 were differentially expressed within the first two hours (Appendix A). These TFs were classified into 38 families. In RIO, the three early response TF families with the largest number were bHLH (15), WRKY (13) and NAC (11) (Figure 5C). While in SN005, they were ERF (22), MYB (17) and HSF (13) (Figure 5C). The differences in the early responsive TFs would result in different downstream gene regulation and final reactions to salt stimulation. This revealed the different stress response patterns between the two genotypes in the early stage under salt stress.

From the expression pattern perspective, we identified 120 TFs that were expressed with similar patterns between genotypes (Appendix A). The top 3 common response TF types belonged to the bHLH (13), MYB-related (10) and HSF (8) families (Figure 5D). Among genotype-specific response TFs, we identified 189 specific response TFs in RIO and 172 in SN005. The striking result was that most MYB-, bZIP-, bHLH- and ERF-type TFs showed different expression patterns between genotypes (Figure 5E,F). Among genotype-specific response TFs, 94 (49.7%) TFs in RIO and 54 (31.9%) TFs in SN005 were early response TFs (Appendix A).

The different response patterns of TFs between genotypes revealed different response mechanisms. According to the gene functional annotation, several tolerant line-specific response TFs were involved in hormone signaling and secondary metabolic processes, consistent with the GO enrichment analysis results (Figure 3 and Figure 4). For example, *Sobic.007G178300* was a tolerant line specific response MYB-type TF that quickly responded at 1 h after salt treatment only in RIO (Appendix A). It is a homologous gene of *AtMYB42* that regulates the synthesis of secondary metabolites and mediates salt tolerance in Arabidopsis [47,48]. *Sobic.001G079500* is a C2H2-type TF and was upregulated at 1 h after salt treatment only in RIO (Appendix A). The homologous gene of *Sobic.001G079500* in Arabidopsis is *ZAT10* (*AT1G27730*). *ZAT10* is an ABA-responsive gene and plays important roles in regulating anthocyanin biosynthesis and abiotic stress resistance in Arabidopsis [49,50]. The different response patterns of these TFs might explain the different reactions between genotypes. These TFs might play crucial roles in salt stress resistance in sorghum.

### 2.8. Validation of DEGs by qRT-PCR

The fold change of the DEGs was verified by qRT-PCR. Twelve DEGs were selected to validate by qRT-PCR, including five enzyme genes in the secondary metabolic pathway (*Sobic.002G368000*, *Sobic.005G137300*, *Sobic.003G230900*, *Sobic.010G071800* and *Sobic.002G366500*), an aquaporin gene (*Sobic.002G125700*), a *JAZ* gene (*Sobic.002G214800*) and five TFs (*Sobic.003G373000*, *Sobic.008G020300*, *Sobic.007G062200*, *Sobic.007G178300* and *Sobic.001G079500*) (Figure 6). The fold change of these DEGs were basically consistent with the RNA-seq results (Appendix A), confirming the reproducibility of the data.

## 3. Discussion

Sorghum is considered to have high salt tolerance ability [51]. However, this ability is variable in different genotypes [16,22]. We evaluated the salt tolerance ability of several sweet sorghum inbred lines and observed large salt tolerance ability variations among genotypes (Appendix A). We further evaluated the salt tolerance ability of two sweet sorghum genotypes (RIO and SN005) in detail because they presented large differences in salt tolerance ability (Figure 1). Our results revealed that both genotypes were inhibited by salt stress in both the germination stage and seedling stage, but RIO was more tolerant than SN005, especially under high NaCl concentrations (200 and 250 mM). The germination rate, relative dry weight, actual PSII efficiency and chlorophyll content of SN005 severely decreased in 250 mM NaCl solution, while RIO was relatively mildly inhibited at such high NaCl concentrations (Figure 1 and Appendix A). These data reflected that the variation of the salt tolerance phenotype in sweet sorghum might appear at high salt concentrations. According to these results, a comparative study of these two genotypes can provide insights into salt tolerance mechanisms in sweet sorghum.

Salt tolerance of plants is a quantitative trait that is controlled by multiple genes [52,53]. Transcriptome analysis can scan the expression pattern of global genes and draw an overview picture to reveal the response process, which can be used to identify the mechanisms and key genes involved in salt tolerance. However, most of the transcriptome studies for salt stress in sorghum only captured the expression level at 1–3 time points [20,22,54]. Considering that the response to salt stress is a continuously dynamic process, our study provided a dynamic global transcript overview based on transcriptome sequencing at 6 time points (Appendix A). Thus, our data can be used to monitor more comprehensive transcriptomic dynamic changes under salt stress in sweet sorghum.

The differential expression and GO enrichment analysis showed that the genes in the secondary metabolite and hormone signaling pathways responded more quickly in the tolerant line than in the sensitive line and restored homeostasis earlier than in the sensitive line (Figure 2 and Figure 3). Furthermore, we performed comparative analysis of gene expression patterns between two genotypes. Based on the gene expression pattern, we divided DEGs into 3 sets: common response, tolerant line-specific response and sensitive line-specific response. Tolerant line-specific response genes were significantly enriched in the categories related to secondary metabolic pathways (Figure 4, Appendix A). Detailed analysis of genes in secondary metabolic pathways and hormone signaling showed that more salt-tolerant genes responded within 2 h in the salt-tolerant line RIO (Appendix A). These results illustrated that an early gene response in secondary metabolic pathways and hormone signaling existed in the salt-tolerant line. This might explain the different salt tolerance abilities between the two genotypes. As previous studies have discovered multiple salt tolerance mechanisms in plants [32], our results revealed the critical role of secondary metabolite and hormone signaling in salt tolerance in sweet sorghum.

Under stress conditions, transcription factors (TFs) play a crucial role in the transduction of stress signal perception to responsive gene expression. The interaction of transcription factors and *cis*-elements in the promoter region acts as a molecular switch to express stress-responsive genes [38,55]. Many transcription factors in different transcription factor families are reported in abiotic stress responses, such as NAC, bZIP, MYB, WRKY, ERF and HSFs [39,40,56,57,58,59]. In our study, more than 300 TFs were expressed differently at different time points in both lines (349 in RIO and 354 in SN005). Approximately half of them were expressed differently in the early stage of treatment (Figure 5). These results revealed the crucial role of transcription factors in the stress response process in sweet sorghum, especially in the early responses. This is consistent with previous studies in other plants [60]. Through comparative analysis of gene expression patterns, we found that the largest number of tolerant line-specific response TFs was MYB-type TFs (Figure 5E). MYB-type TFs have been proven to be critical in the biosynthesis of secondary metabolites and abiotic stress responses in plants [61]. The different expression patterns of MYB-type TFs might cause the different response patterns of genes in secondary metabolite biosynthesis between genotypes.

Taken together, we described the dynamic transcriptome regulatory network in two sweet sorghum genotypes, the salt-tolerant genotype RIO and the salt-sensitive genotype SN005, under salt treatment. By comparative transcriptome analysis, we found that an early response, especially in secondary metabolism and hormone signaling, might be essential for salt tolerance in sweet sorghum. Our results provide further insight into the dynamic and complex regulatory networks of salt tolerance in sorghum.

## 4. Materials and Methods

### 4.1. Plant Materials, Growth Conditions and Treatments

For the germination experiment, 90 seeds (3 repeats, 30 seeds per repeat) for each genotype were sterilised with 15% bleach (NaClO) and germinated in a 35 mm plastic petri dish containing 3 pieces of filter paper under 4 concentrations of NaCl: 0 mM, 100 mM, 200 mM and 300 mM. The number of germinated and non-germinated seeds was counted after 7 days.

Two sorghum genotypes, RIO and SN005, were selected for future study. After being soaked with water at 28 °C for 12 h, plump seeds were selected and sown in pots (10 × 10 × 5 cm, ten seeds in each pot) filled with a 1:1 mix of vermiculite: nutrient soil and irrigated with water. After germination, the seedlings were cultured in controlled greenhouse conditions of 28/22 °C (day/night) at a light intensity of 600 μmol m^−2^ s^−1^ (16-h-light/8-h-night) and 40–50% relative humidity. To identify salt tolerance phenotypes, the 3-leaf stage seedlings were irrigated with water and 3 concentrations of NaCl solution: 150, 200 and 250 mM. The evaporated water was replenished every day. Phenotypic photos were taken after 15 days. Relative dry weight, actual PSII efficiency (ΦPSII) and relative chlorophyll content were measured at 7 days after treatment. The leaves of 15 sorghum plants (3 repeats, 5 plants per repeat) were harvested and dried at 70 °C for 1 week. Relative dry weight was defined as relative dry weight = treatment/control. Pulse-amplitude modulation (PAM) fluorometry and relative chlorophyll content (SPAD) were carried out using a MultispeQ-Beta device and PhotosynQ platform software (Kuhlgert et al., 2016). For RNA-seq treatment, the seedlings of the 2 sorghum genotypes grew under normal conditions until the 4-leaf stage. For salt treatment, plants were irrigated with 200 mM NaCl solution. The shoot tissue from two genotypes in three replicates was collected and flash frozen in liquid nitrogen before salt treatment and 1 h, 2 h, 6 h, 12 h and 24 h after salt treatment.

### 4.2. Total RNA Extraction, Library Construction and Illumina Sequencing

Total RNA was isolated using the Total Plant RNA Extraction Kit (Karroten, Beijing, China) following the manufacturer’s protocols. RNA degradation and contamination were monitored on 1% agarose gels. RNA purity was checked using a NanoPhotometer^®^ spectrophotometer (IMPLEN, Calabasas, CA, USA). RNA concentration was measured using the Qubit^®^ RNA Assay Kit in a Qubit^®^ 2.0 Fluorometer (Life Technologies, Carlsbad, CA, USA). RNA integrity was assessed using the RNA Nano 6000 Assay Kit of the Bioanalyzer 2100 system (Agilent Technologies, Santa Clara, CA, USA). A total amount of 5 μg RNA per sample was used as input material for the RNA sample preparations. Sequencing libraries were generated using the NEBNext^®^ Ultra™ RNA Library Prep Kit for Illumina^®^ (New England Biolabs, Inc., Ipswich, MA, USA) following the manufacturer’s recommendations. The library was sequenced on an Illumina HiSeq^TM^ 2500 platform.

### 4.3. Differentially Expressed Gene Detection and Functional Analysis

The BTx623 genome dataset was downloaded from the Phytozome website (https://phytozome.jgi.doe.gov; accessed on 8 June 2019). RNA-seq clean reads were aligned to the reference genome using TopHat version 2.1.1, allowing a 2-segment mismatch [62,63]. HTseq-count version 0.13.5 with the default parameter was used to count the read numbers mapped to each gene [64]. Only the unique mapped reads were used to estimate read counts for each gene. Then, the fragments per kilobase of transcript per million fragments mapped (FPKM) value for each gene was calculated by an in-house script based on the count table of HTseq-count output. Differential expression analysis was performed by the edgeR R package (version 3.24.3) [65]. Genes with fold changes ≥ 2 and a false discovery rate (FDR)-adjusted *p* value < 0.01 were assigned as differentially expressed.

GO term enrichment analysis of DEGs was performed by GO::TermFinder version 0.86 [66]. GO terms with a *p* value less than 0.05 and FDR (adjusted *p* value) less than 0.1 were considered significantly enriched in our study. The transcription factor information of sorghum was downloaded from PlantTFDB (http://planttfdb.cbi.pku.edu.cn; accessed on 10 December 2019) [67].

### 4.4. Gene Expression Pattern Analysis

Gene expression pattern analysis was performed by the WGCNA package in R [33]. To filter the low-abundance or non-varying genes, DEGs with mean FPKM ≥ 1 at least at 2 time points were selected in the 2 genotypes. The time-course FPKM values of selected genes from two genotypes were labeled by the source and combined as input. The FPKM data was imported into WGCNA. The co-expression modules were calculated by the blockwiseModules function in the WGCNA package with soft thresholding 26, minModuleSize 30 and mergeCutHeight 0.25.

### 4.5. Quantitative Real-Time PCR Analysis

A Bio–Rad iQ5 thermocycler (Bio-Rad Laboratories, Hercules, CA, USA) was used to perform quantitative real-time PCR analysis. Twenty-one DEGs were randomly selected, and quantitative real-time PCR was performed at different expression time points to verify the RNA-seq results. Primers for these genes were designed by the Primer-BLAST tool on the NCBI website (https://www.ncbi.nlm.nih.gov/tools/primer-blast/; accessed on 10 July 2020) (Appendix A). First-strand cDNA was synthesised by a FastQuany RT Kit with gDNase (TIANGEN, Beijing, China). The 20-μL reaction solution contained 1 ng cDNA, 0.3 μM gene-specific primers and 10 μL 2 × Talent qPCR PreMix (TIANGEN). Three replicate qPCRs were run per cDNA sample. The qPCR reactions were performed under the following thermal cycling conditions: 95 °C for 3 min and 40 cycles at 95 °C for 15 s and 60 °C for 15 s. Three biological replicates of each sample and three technical replicates of each biological replicate were used for qRT-PCR analysis. Relative quantification analyses were performed using the 2^−^^ΔΔCT^ method [68]. *SbPP2A* was used as an endogenous control gene [69]. The ΔCT was calculated by the difference in threshold cycle (CT) between the target and endogenous control gene. The ΔΔCT was calculated by the difference in ΔCT of target genes between each time point and control. The fold change was calculated by 2^−^^ΔΔCT^ [68].

### 4.6. Statistical Analysis

A pairwise comparison test of phenotypes (Figure 1B–D) and relative expression levels (Figure 6) among samples was performed by Duncan’s multiple range test at the 0.05 significance level. All tests were performed with IBM SPSS Statistics for Windows version 19.0 (IBM Corp, Armonk, NY, USA).

## Figures and Tables

**Figure 1 ijms-23-02272-f001:**
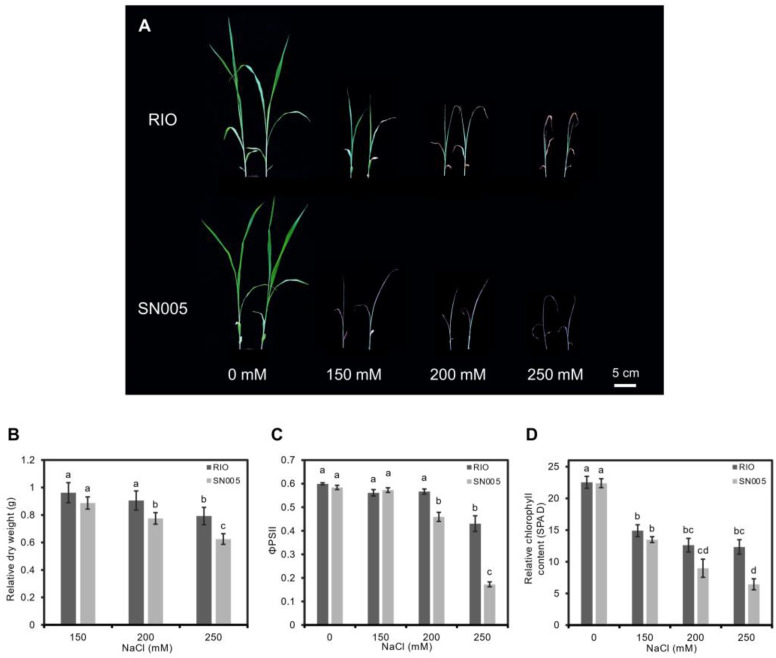
Phenotypic and physiological responses to salt stress in two genotypes with different concentrations of NaCl. (**A**) Phenotype of RIO and SN005 seedlings at 15 days after treatment. (**B**) Relative dry weight at 7 days after treatment. The relative rate was defined as relative dry weight = treatment/control. (**C**) Actual PSII efficiency (ΦPSII) at 7 days after treatment. (**D**) Relative chlorophyll content (SPAD) at 7 days after treatment. Values are means ± SE for each measurement. Bars with different lowercase letters are significantly different at *p* < 0.05 (Duncan’s multiple range test).

**Figure 2 ijms-23-02272-f002:**
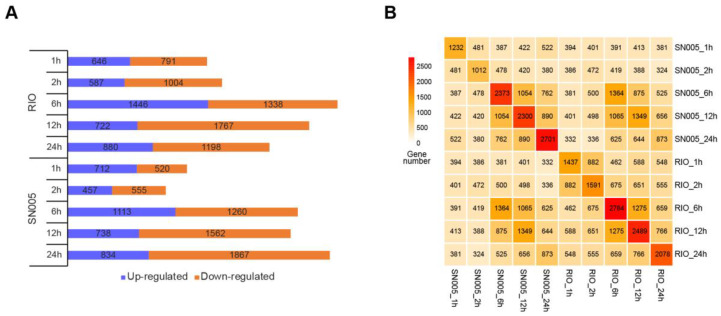
Overview of differentially expressed genes (DEGs) in the tolerant line RIO and the sensitive line SN005 after salt treatment. (**A**) Number of upregulated and downregulated DEGs in the two genotypes at each time point. (**B**) The number of overlapping DEGs between different genotypes at different time points. The colors of heatmap cells indicate the number of DEGs. The number in heatmap cells indicates the number of overlapping DEGs between DEGs set in the horizontal axis and vertical axis.

**Figure 3 ijms-23-02272-f003:**
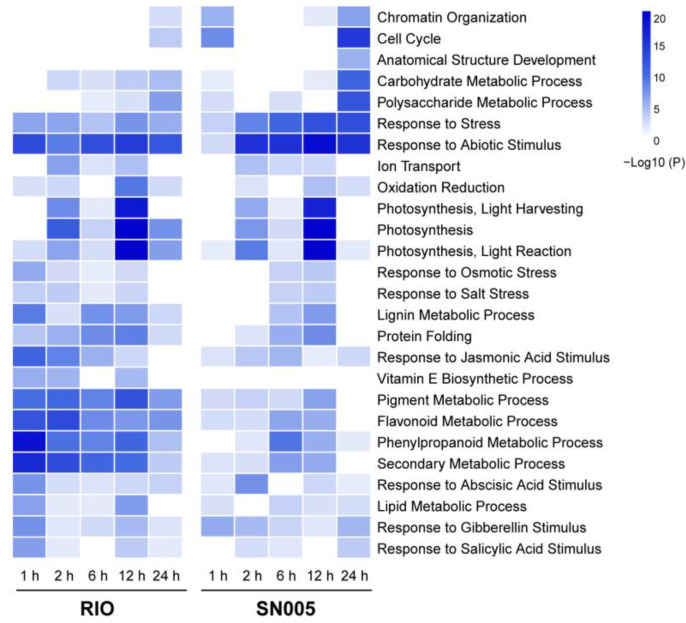
Significantly enriched GO terms of DEGs at each time point in the tolerant line RIO and the sensitive line SN005. The color of heatmap cells represents the log-transformed *p* value of GO enrichment analysis.

**Figure 4 ijms-23-02272-f004:**
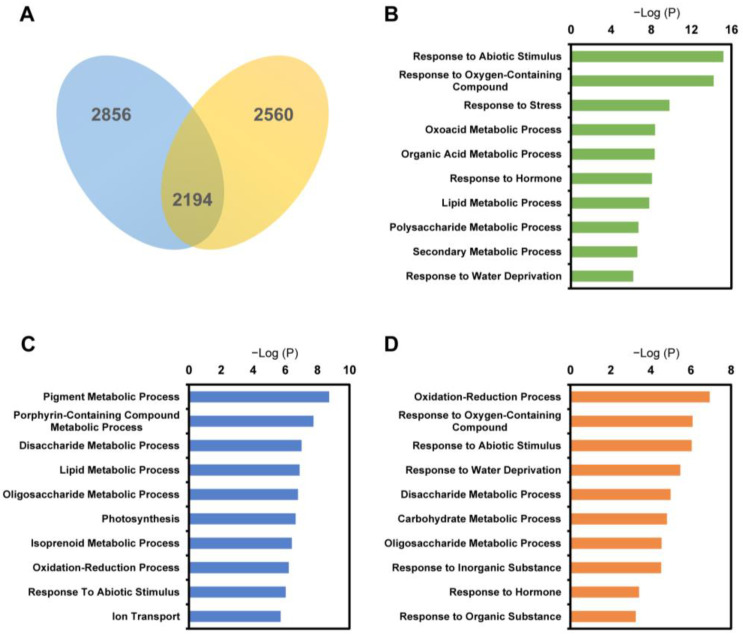
Gene number and significantly enriched GO terms of expression pattern cluster gene sets. Expression pattern cluster analysis divided DEGs into three gene sets: common response DEGs, RIO-specific response DEGs and SN005-specific response DEGs. (**A**) Venn diagram showing the gene number for three gene sets. (**B**) Top 10 significantly enriched GO terms for common response DEGs, (**C**) RIO-specific response DEGs, and (**D**) SN005-specific response DEGs. The GO terms with *p* < 0.01 were considered significantly enriched.

**Figure 5 ijms-23-02272-f005:**
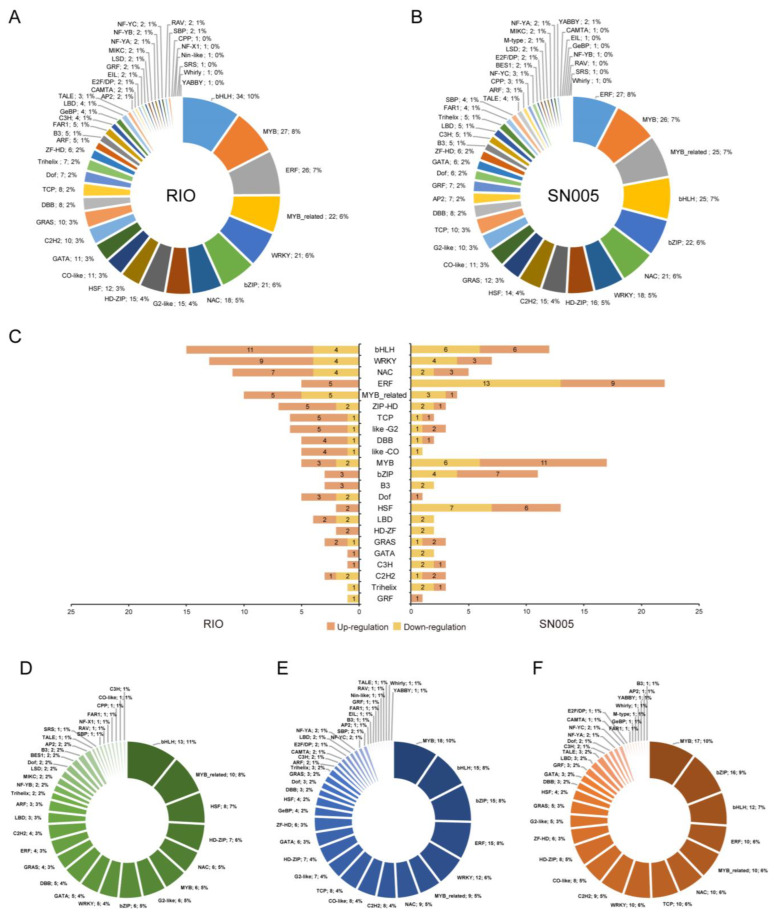
Families of differentially expressed transcription factors in the two genotypes. Distribution in families of differentially expressed transcription factors in the tolerant line RIO (**A**) and the sensitive line SN005 (**B**) at all time points. (**C**) Distribution of up- and downregulated transcription factors in the two genotypes at the early stage (1 and 2 h). Distribution of families of transcription factors in common response gene sets (**D**), RIO-specific response gene sets (**E**), and SN005-specific response gene sets (**F**). Pie charts show the percentage of genes in each transcription factor family.

**Figure 6 ijms-23-02272-f006:**
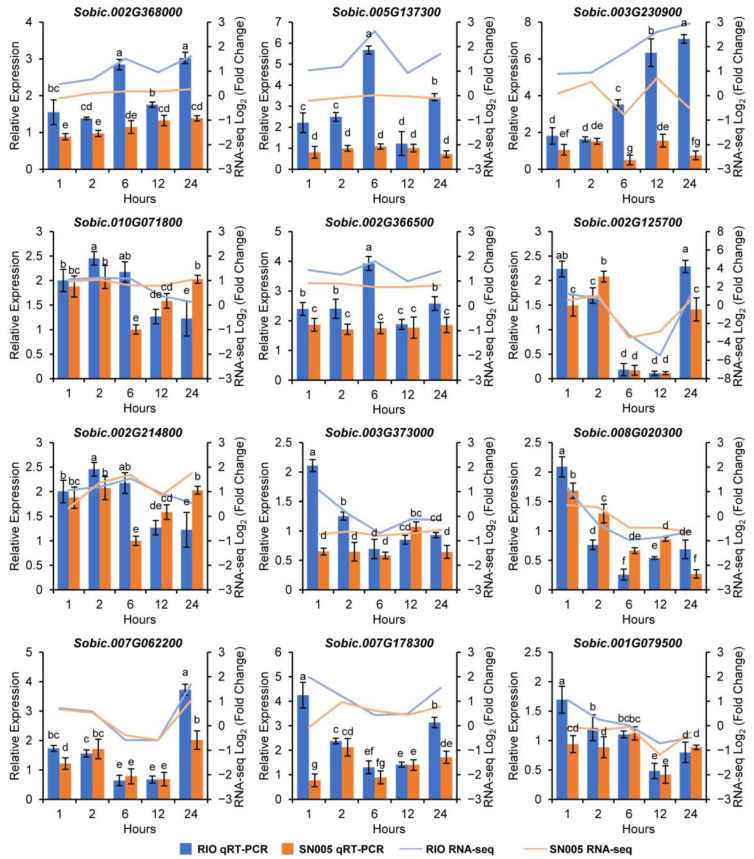
Quantitative RT-PCR validation of 12 DEGs. Three biological replicates of each sample were used for qRT-PCR analysis. Expression of DEGs was normalised to the endogenous control *SbPP2A* gene (ΔCT_gene_ = CT_gene_ − CT*_SbPP2A_*). The ΔΔCT of each biological replicate was calculated by the difference of ΔCT between each replicate and the mean ΔCT of CKs. The relative expression level was calculated by 2^−^^ΔΔCT^. Error bars represent the standard deviation of relative expression level from three biological replicates. Bars with different lowercase letters are significantly different at *p* < 0.05 (Duncan’s multiple range test). The lines show the Log_2_ fold change expression of the DEGs from RNA-seq data.

## Data Availability

The raw sequence data reported in this paper have been deposited in the Genome Sequence Archive [70] in BIG Data Center [71], Beijing Institute of Genomics (BIG), Chinese Academy of Sciences, under accession numbers PRJCA000350, which are publicly accessible at http://bigd.big.ac.cn/gsa; accessed on 6 January 2022.

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
