# Peer review of "Comparative Transcriptome Analysis of Two Sweet Sorghum Genotypes with Different Salt Tolerance Abilities to Reveal the Mechanism of Salt Tolerance"

_ijms, 2022, doi:10.3390/ijms23042272_

Round 1

Reviewer 1 Report

Chen et al. performed RNA-seq study to perform comparative analysis of gene expression in salt-tolerant and salt-sensitive sweet sorghum lines. While the study provides novel information on salt tolerance, it is important that this study provide the genes potentially involved in the abiotic stress tolerance pathway in sweet sorghum in the form of pathway analysis if possible. I hope the authors will find the following comments useful in improving the manuscript and clarifying the message that the study is aiming to convey.

Major comments:

  1. There were few lines with similar germination rate in S1 as compared to Rio. How and why was Rio selected as salt tolerant in this study? Was there any significant difference between Rio and other lines?
  2. How were reads mapping to multiple sites in the genome handled?
  3. How was statistical analyses performed in the study? The whole section is completely missing from the manuscript.
  4. Instead of randomly selecting DEGs from RNA-seq for qRT-PCR validation, it is recommended that DEGs that are identified to be involved in salt tolerance in the study be validated for meaningful conclusions. Report the validation results in the main body rather than supplementary data.
  5. How were the time points of sampling for RNA-seq determined in the study given that expression of genes are differential after one hour and also after 24 hour?
  6. As authors mentioned in the introduction there are several other studies done before in sweet sorghum performing comparative transcriptome studies for salt stress. However, these studies have not been addressed and cited in this study. It is recommended that authors perform rigorous study of previous literature and include them in the context of this study to develop the story.

Minor comments:

Line 43: originated from

Line 44: Be specific on what kind of environment you are referring to “environment of Africa”

Line 74: Include reference to the data being presented

Line 88: compared to

Line 134: Explain “… different physiological statuses between genotypes …”

Line 166: Include the names of two genotypes in the captions and wherever applicable. Figures and tables should be self-explanatory.

Line 394: Use complete sentences: “Replenish the evaporated water every day”

Line 448: Explain “comparative CT method”

Author Response

Response to Reviewer 1’ comments

Reviewer1

Chen et al. performed RNA-seq study to perform comparative analysis of gene expression in salt-tolerant and salt-sensitive sweet sorghum lines. While the study provides novel information on salt tolerance, it is important that this study provide the genes potentially involved in the abiotic stress tolerance pathway in sweet sorghum in the form of pathway analysis if possible. I hope the authors will find the following comments useful in improving the manuscript and clarifying the message that the study is aiming to convey.

Comment #1

Major comments:

  1. (1) There were few lines with similar germination rate in S1 as compared to Rio. How and why was Rio selected as salt tolerant in this study? (2) Was there any significant difference between Rio and other lines?

Response:

(1) We chosen Rio in this study because it is a widely used sweet sorghum line in previous studies. (2) The germination rates of SN005, SN010 and YC001 were significantly different from that of Rio at different salt concentrations (100 mM, 200 mM and 300 mM). The germination rates of SN009 and M81-E were significantly different from that of Rio at 100 mM salt treatment. There were no significant differences between Rio and other lines at other salt concentrations. We have added this in the figure legend of S1.

Comment #2

  1. How were reads mapping to multiple sites in the genome handled?

Response:

In our study, HTseq-count package was used to count the reads mapping to each gene with the default parameter (--nonunique none). According to this parameter the reads mapping to multiple sites were excluded. We have added this in Materials and Methods.

Comment #3

  1. How was statistical analyses performed in the study? The whole section is completely missing from the manuscript.

Response:

Statistical Analysis section has been added in Materials and Methods to explain how statistical analyses was performed in the study (line 481).

Comment #4

  1. Instead of randomly selecting DEGs from RNA-seq for qRT-PCR validation, it is recommended that DEGs that are identified to be involved in salt tolerance in the study be validated for meaningful conclusions. Report the validation results in the main body rather than supplementary data.

Response:

We have presented the validation results in a new section and a new figure (section 2.8 and Figure 6) on main body.

Comment #5

  1. How were the time points of sampling for RNA-seq determined in the study given that expression of genes are differential after one hour and also after 24 hour?

Response:

Previous studies have shown that plants present different physiological responses as time goes on after salt treatment, including osmotic stress, ion toxicity and oxidative stress (Innovation (N Y). 2020;1(1):100017). Most of these studies were performed within 24 hours after the treatments. Therefore, we chosen four time points (1, 2, 6, 12 and 24 hours after salt treatment) to explore the gene expression pattern.

Comment #6

  1. As authors mentioned in the introduction there are several other studies done before in sweet sorghum performing comparative transcriptome studies for salt stress. However, these studies have not been addressed and cited in this study. It is recommended that authors perform rigorous study of previous literature and include them in the context of this study to develop the story.

Response:

We have added the references (line62).

Comment #7

Minor comments:

Line 43: originated from

Response:

Done.

Comment #8

Line 44: Be specific on what kind of environment you are referring to “environment of Africa”

Response:

Done. “environment of Africa”-> “the hot semi-arid environment of Africa”.

Comment #9

Line 74: Include reference to the data being presented

Line 88: compared to

Response:

Done.

Comment #10

Line 134: Explain “… different physiological statuses between genotypes …”

Response:

Here based on the different of overlapping DEGs, we just speculated that the different physiological statuses may be different between genotypes. We have rewritten the sentence to clarify. “Fewer overlapping DEGs also appeared at 24 h (873), suggesting that the physiological statuses may be different between genotypes at a later stage.”

Comment #11

Line 166: Include the names of two genotypes in the captions and wherever applicable. Figures and tables should be self-explanatory.

Line 394: Use complete sentences: “Replenish the evaporated water every day”

Response:

Done.

Comment #12

Line 448: Explain “comparative CT method”

Response:

Explained in Methods (line476). “Relative quantification analyses were using the 2-ΔΔCT method. SbPP2A was used as a reference gene. The ΔCT was calculated by the difference in threshold cycle (CT) between the target and reference genes. The ΔΔCT was calculated by the difference in ΔCT of target genes between samples. The fold changes were calculated by 2-ΔΔCT.”

Reviewer 2 Report

"most of these studies focused on one or two time points" => give more references
"DEGs with mean FPKM ≥ 1 at least at two time points were selected" => explain why these criteria were taken
"did you used signed or unsigned network"?
in "figure 4" please add line indicated significance level
elorborate more on the transcription factors that are different between RIO/SN005 (such as ERF)..e.g. which genes .. to which pathway do they belong to etc.

Author Response

Response to Reviewer 2’ comments

Reviewer 2

Comment #1

"most of these studies focused on one or two time points" => give more references

Response:

Done.

Comment #2
"DEGs with mean FPKM ≥ 1 at least at two time points were selected" => explain why these criteria were taken

Response:

These criteria were used to filter the low-abundance or non-varying genes, which are meaningless in WGCNA analysis. To filter the low-abundance genes, the common filter criteria is the FPKM ≥ 1 in RNA-seq studies. Moreover, we selected the genes with FPKM ≥ 1 at least at two time points to filter the non-varying genes. We have modified the description by replacing that sentence with “To filter the low-abundance or non-varying genes, DEGs with mean FPKM ≥ 1 at least at two time points were selected.” (line458)

Comment #3
"did you used signed or unsigned network"?

Response:

In our study, we used the unsigned weight to construct co-expression network for finding clusters of highly correlated genes, because we concern about the strength of relatedness between two genes.

Comment #4
in "figure 4" please add line indicated significance level

Response:

Done. We have added a sentence at figure legend (line213) “The GO terms with p < 0.01 were considered significantly enriched.”

Comment #5
elorborate more on the transcription factors that are different between RIO/SN005 (such as ERF)..e.g. which genes .. to which pathway do they belong to etc.

Response:

Done. We have illustrated more di

Round 2

Reviewer 1 Report

I would like to thank Chen et al. for considering the reviewers’ comments and making changes to the manuscript. Although authors have mentioned changes in the rebuttal, the changes are not made in the manuscript. It is highly recommended that authors reflect the changes in the manuscript and with tracked changes.

  1. What pipeline was used to do mapping?
  2. Explain in detail how and on what analyses you used t-test and pairwise comparison test in statistical analyses section. How were the means of the three replicates averaged to generate gene expression histrograms in figure 6.
  3. Change the y-axis label to relative expression
  4. Include the RNAseq expression values from transcriptome data in each histograms similar to figure 5 in this paper: https://bmcplantbiol.biomedcentral.com/articles/10.1186/s12870-020-02742-4. This will give a good visual comparison of transcriptome and qRT-PCR data.
  5. Although authors indicated that newer references were added in line 62, there are no references mentioned in the line. Adding recent findings will help in connecting the research to the current information. Consider following if appropriate:

Yang, Z., Zheng, H., Wei, X., Song, J., Wang, B., & Sui, N. (2018). Transcriptome analysis of sweet Sorghum inbred lines differing in salt tolerance provides novel insights into salt exclusion by roots. Plant and soil, 430(1), 423-439.

Cui, J., Ren, G., Qiao, H., Xiang, X., Huang, L., & Chang, J. (2018). Comparative transcriptome analysis of seedling stage of two sorghum cultivars under salt stress. Journal of plant growth regulation, 37(3), 986-998.

Zheng, H., Sun, X., Li, J., Song, Y., Song, J., Wang, F., ... & Sui, N. (2021). Analysis of N6-methyladenosine reveals a new important mechanism regulating the salt tolerance of sweet sorghum. Plant Science, 304, 110801.

Author Response

Authors’ Response to Review Comments

I would like to thank Chen et al. for considering the reviewers’ comments and making changes to the manuscript. Although authors have mentioned changes in the rebuttal, the changes are not made in the manuscript. It is highly recommended that authors reflect the changes in the manuscript and with tracked changes.

Response:

Thank you again for your insightful comments and suggestions. We have re-uploaded the revised manuscript with tracked changes.

Comment #1

  1. What pipeline was used to do mapping?

Response:

In our study, TopHat v2.1.1, allowing a 2-segment mismatch, was used for mapping clean reads to the reference genome. We have added this in methods.

Comment #2

  1. (1) Explain in detail how and on what analyses you used t-test and pairwise comparison test in statistical analyses section. (2) How were the means of the three replicates averaged to generate gene expression histrograms in figure 6.

Response:

(1) Pairwise comparison test of phenotypes (Figure 1b-d) and relative expression levels (Figure 6) among samples were performed by Duncan’s multiple range test at the 0.05 significance level in SPSS. We have added this in statistical analyses section.

(2) Relative quantification analyses were performed using the 2-ΔΔCT method. Three biological replicates of each sample and three technical replicates of each biological replicate were used for qRT-PCR analysis. The CT values of each biological replicate was calculated by the mean value of three technical replicates. The ΔCT of each biological replicate was calculated by the difference of CT between the target and reference genes. The ΔΔCT of each biological replicate was calculated by the difference of ΔCT between each replicate and the mean ΔCT of control samples. The relative expression level between treatment and control was calculated by 2-ΔΔCT. The means of relative expression level from three replicates were averaged to generate relative expression histograms. Error bars represent the standard deviation of relative expression level from three biological replicates. We have added this in legend of figure 6 and methods.

Comment #3

  1. Change the y-axis label to relative expression
  2. Include the RNAseq expression values from transcriptome data in each histograms similar to figure 5 in this paper: https://bmcplantbiol.biomedcentral.com/articles/10.1186/s12870-020-02742-4. This will give a good visual comparison of transcriptome and qRT-PCR data.

Response:

Done.

Comment #4

  1. Although authors indicated that newer references were added in line 62, there are no references mentioned in the line. Adding recent findings will help in connecting the research to the current information. Consider following if appropriate:

Yang, Z., Zheng, H., Wei, X., Song, J., Wang, B., & Sui, N. (2018). Transcriptome analysis of sweet Sorghum inbred lines differing in salt tolerance provides novel insights into salt exclusion by roots. Plant and soil430(1), 423-439.

Cui, J., Ren, G., Qiao, H., Xiang, X., Huang, L., & Chang, J. (2018). Comparative transcriptome analysis of seedling stage of two sorghum cultivars under salt stress. Journal of plant growth regulation37(3), 986-998.

Zheng, H., Sun, X., Li, J., Song, Y., Song, J., Wang, F., ... & Sui, N. (2021). Analysis of N6-methyladenosine reveals a new important mechanism regulating the salt tolerance of sweet sorghum. Plant Science304, 110801.

Response:

Thank you for commending valuable literatures. We have added such references at line 62.

Round 3

Reviewer 1 Report

I thank Chen et al. for addressing reviewers' comments and making required changes in the manuscript. Please pay attention to minor language errors. I do not have any additional comments.